# Deep Set Prediction Networks

**Yan Zhang**
University of Southampton
Southampton, UK
yz5n12@ecs.soton.ac.uk

**Jonathon Hare**
University of Southampton
Southampton, UK
jsh2@ecs.soton.ac.uk

**Adam Prügel-Bennett**
University of Southampton
Southampton, UK
apb@ecs.soton.ac.uk

## Abstract

Current approaches for predicting sets from feature vectors ignore the unordered nature of sets and suffer from discontinuity issues as a result. We propose a general model for predicting sets that properly respects the structure of sets and avoids this problem. With a single feature vector as input, we show that our model is able to auto-encode point sets, predict the set of bounding boxes of objects in an image, and predict the set of attributes of these objects.

## 1 Introduction

You are given a rotation angle and your task is to draw the four corner points of a square that is rotated by that amount. This is a structured prediction task where the output is a *set*, since there is no inherent ordering to the four points. Such sets are a natural representation for many kinds of data, ranging from the set of points in a point cloud, to the set of objects in an image (object detection), to the set of nodes in a molecular graph (molecular generation). Yet, existing machine learning models often struggle to solve even the simple square prediction task [30].

The main difficulty in predicting sets comes from the ability to permute the elements in a set freely, which means that there are $n!$ equally good solutions for a set of size $n$. Models that do not take this set structure into account properly (such as MLPs or RNNs) result in discontinuities, which is the reason why they struggle to solve simple toy set prediction tasks [30]. We give background on what the problem is in section 2.

How can we build a model that properly respects the set structure of the problem so that we can predict sets without running into discontinuity issues? In this paper, we aim to address this question. Concretely, we contribute the following:

1. We propose a model (section 3, Algorithm 1) that can predict a set from a feature vector (vector-to-set) while properly taking the structure of sets into account. We explain what properties we make use of that enables this. Our model uses backpropagation through a set encoder to decode a set and works for variable-size sets. The model is applicable to a wide variety of set prediction tasks since it only requires a feature vector as input.

2. We evaluate our model on several set prediction datasets (section 5). First, we demonstrate that the auto-encoder version of our model is sound on a set version of MNIST. Next, we use the CLEVR dataset to show that this works for general set prediction tasks. We predict the set of bounding boxes of objects in an image and we predict the set of object attributes in an image, both from a single feature vector. Our model is a completely different approach to usual anchor-based object detectors because we pose the task as a set prediction problem, which does not need complicated post-processing techniques such as non-maximum suppression.

## 2  Background

**Representation**   We are interested in sets of feature vectors with the feature vector describing properties of the element, for example the 2d position of a point in a point cloud. A set of size $n$ wherein each feature vector has dimensionality $d$ is represented as a matrix $\boldsymbol{Y} \in \mathbb{R}^{d \times n}$ with the elements as columns in an arbitrary order, $\boldsymbol{Y} = [\boldsymbol{y}_1, \ldots, \boldsymbol{y}_n]$. To properly treat this as a set, it is important to only apply operations with certain properties to it [29]: *permutation-invariance* or *permutation-equivariance*. In other words, operations on sets should not rely on the arbitrary ordering of the elements.

Set encoders (which turn such sets into feature vectors) are usually built by composing permutation-equivariant operations with a permutation-invariant operation at the end. A simple example is the model in [29]: $f(\boldsymbol{Y}) = \sum_i g(\boldsymbol{y}_i)$ where $g$ is a neural network. Because $g$ is applied to every element individually, it does not rely on the arbitrary order of the elements. We can think of this as turning the set $\{\boldsymbol{y}_i\}_{i=1}^n$ into $\{g(\boldsymbol{y}_i)\}_{i=1}^n$. This is permutation-equivariant because changing the order of elements in the input set affects the output set in a predictable way. Next, the set is summed to produce a single feature vector. Since summing is commutative, the output is the same regardless of what order the elements are in. In other words, summing is permutation-invariant. This gives us an encoder that produces the same feature vector regardless of the arbitrary order the set elements were stored in.

**Loss**   In set prediction tasks, we need to compute a loss between a predicted set $\hat{\boldsymbol{Y}} = [\hat{\boldsymbol{y}}_1, \ldots, \hat{\boldsymbol{y}}_n]$ and the target set $\boldsymbol{Y}$. The main problem is that the elements of each set are in an arbitrary order, so we cannot simply compute a pointwise distance. The usual solution to this is an assignment mechanism that matches up elements from one set to the other set. This gives us a loss function that is permutation-invariant in both its arguments.

One such loss is the $O(n^2)$ Chamfer loss, which matches up every element of $\hat{\boldsymbol{Y}}$ to the closest element in $\boldsymbol{Y}$ and vice versa:

$$L_{\text{cha}}(\hat{\boldsymbol{Y}}, \boldsymbol{Y}) = \sum_i \min_j ||\hat{\boldsymbol{y}}_i - \boldsymbol{y}_j||^2 + \sum_j \min_i ||\hat{\boldsymbol{y}}_i - \boldsymbol{y}_j||^2 \qquad (1)$$

Note that this does not work well for multi-sets: the loss between $[\boldsymbol{a}, \boldsymbol{a}, \boldsymbol{b}]$, $[\boldsymbol{a}, \boldsymbol{b}, \boldsymbol{b}]$ is 0. A more sophisticated loss that does not have this problem involves the linear assignment problem with the pairwise losses as assignment costs:

$$L_{\text{hun}}(\hat{\boldsymbol{Y}}, \boldsymbol{Y}) = \min_{\pi \in \Pi} ||\hat{\boldsymbol{y}}_i - \boldsymbol{y}_{\pi(i)}||^2 \qquad (2)$$

where $\Pi$ is the space of permutations, which can be solved with the Hungarian algorithm in $O(n^3)$ time. This has the benefit that every element in one set is associated to exactly one element in the other set, which is not the case for the Chamfer loss.

**Responsibility problem**   A widely-used approach is to simply ignore the set structure of the problem. A feature vector can be mapped to a set $\hat{\boldsymbol{Y}}$ by using an MLP that takes the vector as input and directly produces $\hat{\boldsymbol{Y}}$ with $d \times n$ outputs. Since the order of elements in $\hat{\boldsymbol{Y}}$ does not matter, it appears reasonable to always produce them in a certain order based on the weights of the MLP.

While this seems like a promising approach, [30] point out that this results in a discontinuity issue: there are points where a small change in set space requires a large change in the neural network outputs. The model needs to "decide" which of its outputs is responsible for producing which element, and this responsibility must be resolved discontinuously.

The intuition behind this is as follows. Consider an MLP that detects the colour of two otherwise identical objects present in an image, so it has two outputs with dimensionality 3 (R, G, B) corresponding to those two colours. We are given an image with a blue and red object, so let us say that output 1 predicts blue and output 2 predicts red; perhaps the weights of output 1 are more attuned to the blue channel and output 2 is more attuned to the red channel. We are given another image with a blue and green object, so it is reasonable for output 1 to again predict blue and output 2 to now predict green. When we now give the model an image with a red and green object, or two red

**Algorithm 1** One forward pass of the set prediction algorithm within the training loop.

1: $\boldsymbol{z} = F(x)$                                                             ▷ encode input with a model
2: $\hat{\boldsymbol{Y}}^{(0)} \leftarrow$ init                                                                      ▷ initialise set
3: **for** t $\leftarrow 1, T$ **do**
4:      $l \leftarrow L_{\text{repr}}(\hat{\boldsymbol{Y}}^{(t-1)}, \boldsymbol{z})$                                        ▷ compute representation loss
5:      $\hat{\boldsymbol{Y}}^{(t)} \leftarrow \hat{\boldsymbol{Y}}^{(t-1)} - \eta \frac{\partial l}{\partial \hat{\boldsymbol{Y}}^{(t-1)}}$                            ▷ gradient descent step on the set
6: **end for**
7: predict $\hat{\boldsymbol{Y}}^{(T)}$
8: $\mathcal{L} = \frac{1}{T} \sum_{t=0}^{T} L_{\text{set}}(\hat{\boldsymbol{Y}}^{(t)}, \boldsymbol{Y}) + \lambda L_{\text{repr}}(\boldsymbol{Y}, \boldsymbol{z})$              ▷ compute loss of outer optimisation

objects, it is unclear which output should be responsible for predicting which object. Output 2 "wants" to predict both red and green, but has to decide between one of them, and output 1 now has to be responsible for the other object while previously being a blue detector. This responsibility must be resolved discontinuously, which makes modeling sets with MLPs difficult [30].

The main problem is that there is a notion of output 1 and output 2 – an ordered output representation – in the first place, which forces the model to give the set an order. Instead, it would be better if the outputs of the model were freely interchangeable – in the same way the elements of the set are interchangeable – to not impose an order on the outputs. This is exactly what our model accomplishes.

## 3   Deep Set Prediction Networks

This section contains our primary contribution: a model for decoding a feature vector into a set of feature vectors. As we have previously established, it is important for the model to properly respect the set structure of the problem to avoid the responsibility problem.

Our main idea is based on the observation that the gradient of a set encoder with respect to the input set is permutation-equivariant (see proof in Appendix A): *to decode a feature vector into a set, we can use gradient descent to find a set that* encodes *to that feature vector*. Since each update of the set using the gradient is permutation-equivariant, we always properly treat it as a set and avoid the responsibility problem. This gives rise to a nested optimisation: an inner loop that changes a set to encode more similarly to the input feature vector, and an outer loop that changes the weights of the encoder to minimise a loss over a dataset.

With this idea in mind, we build up models of increasing usefulness for predicting sets. We start with the simplest case of auto-encoding fixed-size sets (subsection 3.1), where a latent representation is decoded back into a set. This is modified to support variable-size sets, which is necessary for most sets encountered in the real-world. Lastly and most importantly, we extend our model to general set prediction tasks where the input no longer needs to be a set (subsection 3.2). This gives us a model that can predict a set of feature vectors from a single feature vector. We give the pseudo-code of this method in Algorithm 1.

### 3.1   Auto-encoding fixed-size sets

In a set auto-encoder, the goal is to turn the input set $\boldsymbol{Y}$ into a small latent space $\boldsymbol{z} = g_{\text{enc}}(\boldsymbol{Y})$ with the encoder $g_{\text{enc}}$ and turn it back into the predicted set $\hat{\boldsymbol{Y}} = g_{\text{dec}}(\boldsymbol{z})$ with the decoder $g_{\text{dec}}$. Using our main idea, we define a *representation loss* and the corresponding decoder as:

$$L_{\text{repr}}(\hat{\boldsymbol{Y}}, \boldsymbol{z}) = ||g_{\text{enc}}(\hat{\boldsymbol{Y}}) - \boldsymbol{z}||^2 \tag{3}$$

$$g_{\text{dec}}(\boldsymbol{z}) = \underset{\hat{\boldsymbol{Y}}}{\arg \min} \, L_{\text{repr}}(\hat{\boldsymbol{Y}}, \boldsymbol{z}) \tag{4}$$

In essence, $L_{\text{repr}}$ compares $\hat{\boldsymbol{Y}}$ to $\boldsymbol{Y}$ in the latent space. To understand what the decoder does, first consider the simple, albeit not very useful case of the identity encoder $g_{\text{enc}}(\boldsymbol{Y}) = \boldsymbol{Y}$. Solving $g_{\text{dec}}(\boldsymbol{z})$ simply means setting $\hat{\boldsymbol{Y}} = \boldsymbol{Y}$, which perfectly reconstructs the input as desired.

When we instead choose $g_{\text{enc}}$ to be a set encoder, the latent representation $\boldsymbol{z}$ is a permutation-invariant feature vector. If this representation is "good", $\hat{\boldsymbol{Y}}$ will only encode to similar latent variables as $\boldsymbol{Y}$ if the two sets themselves are similar. Thus, the minimisation in Equation 4 should still produce a set $\hat{\boldsymbol{Y}}$ that is the same (up to permutation) as $\boldsymbol{Y}$, except this has now been achieved with $\boldsymbol{z}$ as a bottleneck.

Since the problem is non-convex when $g_{\text{enc}}$ is a neural network, it is infeasible to solve Equation 4 exactly. Instead, we perform gradient descent to approximate a solution. Starting from some initial set $\hat{\boldsymbol{Y}}^{(0)}$, gradient descent is performed for a fixed number of steps $T$ with the update rule:

$$\hat{\boldsymbol{Y}}^{(t+1)} = \hat{\boldsymbol{Y}}^{(t)} - \eta \cdot \frac{\partial L_{\text{repr}}(\hat{\boldsymbol{Y}}^{(t)}, \boldsymbol{z})}{\partial \hat{\boldsymbol{Y}}^{(t)}} \tag{5}$$

with $\eta$ as the learning rate and the prediction being the final state, $g_{\text{dec}}(\boldsymbol{z}) = \hat{\boldsymbol{Y}}^{(T)}$. This is the aforementioned inner optimisation loop. In practice, we let $\hat{\boldsymbol{Y}}^{(0)}$ be a learnable $\mathbb{R}^{d \times n}$ matrix which is part of the neural network parameters.

To obtain a good representation $\boldsymbol{z}$, we still have to train the weights of $g_{\text{enc}}$. For this, we compute the auto-encoder objective $L_{\text{set}}(\hat{\boldsymbol{Y}}^{(T)}, \boldsymbol{Y})$ – with $L_{\text{set}} = L_{\text{cha}}$ or $L_{\text{hun}}$ – and differentiate with respect to the weights as usual, backpropagating through the steps of the inner optimisation. This is the aforementioned outer optimisation loop.

In summary, each forward pass of our auto-encoder first encodes the input set to a latent representation as normal. To decode this back into a set, gradient descent is performed on an initial guess with the aim to obtain a set that encodes to the same latent representation as the input. The same set encoder is used in the encoding and decoding stages.

**Variable-size sets**  To extend this from fixed- to variable-size sets, we make a few modifications to this algorithm. First, we pad all sets to a fixed maximum size to allow for efficient batch computation. We then concatenate an additional mask feature $m_i$ to each set element $\hat{\boldsymbol{y}}_i$ that indicates whether it is a regular element ($m_i = 1$) or padding ($m_i = 0$). With this modification to $\hat{\boldsymbol{Y}}$, we can optimise the masks in the same way as the set elements are optimised. To ensure that masks stay in the valid range between 0 and 1, we simply clamp values above 1 to 1 and values below 0 to 0 after each gradient descent step. This performed better than using a sigmoid in our initial experiments, possibly because it allows exact 0s and 1s to be recovered.

### 3.2  Predicting sets from a feature vector

In our auto-encoder, we used an encoder to produce both the latent representation as well as to decode the set. This is no longer possible in the general set prediction setup, since the target representation $\boldsymbol{z}$ can come from a separate model (for example an image encoder $F$ encoding an image $\boldsymbol{x}$), so there is no longer a set encoder in the model.

When naïvely using $\boldsymbol{z} = F(\boldsymbol{x})$ as input to our decoder, our decoding process is unable to predict sets correctly from it. Because the set encoder is no longer shared in our set decoder, there is no guarantee that optimising $g_{\text{enc}}(\hat{\boldsymbol{Y}})$ to match $\boldsymbol{z}$ converges towards $\boldsymbol{Y}$ (or a permutation thereof). To fix this, we simply add a term to the loss of the outer optimisation that encourages $g_{\text{enc}}(\boldsymbol{Y}) \approx \boldsymbol{z}$ again. In other words, the target set should have a very low representation loss itself. This gives us an additional $L_{\text{repr}}$ term in the loss function of the outer optimisation for supervised learning:

$$\mathcal{L} = L_{\text{set}}(\hat{\boldsymbol{Y}}, \boldsymbol{Y}) + \lambda L_{\text{repr}}(\boldsymbol{Y}, \boldsymbol{z}) \tag{6}$$

with $L_{\text{set}}$ again being either $L_{\text{cha}}$ or $L_{\text{hun}}$. With this, minimising $L_{\text{repr}}(\hat{\boldsymbol{Y}}, \boldsymbol{z})$ in the inner optimisation will converge towards $\boldsymbol{Y}$. The additional term is not necessary in the pure auto-encoder because $\boldsymbol{z} = g_{\text{enc}}(\boldsymbol{Y})$, so $L_{\text{repr}}(\boldsymbol{Y}, \boldsymbol{z})$ is always 0 already.

**Practical tricks**  For the outer optimisation, we can compute the set loss for not only $\hat{\boldsymbol{Y}}^{(T)}$, but all $\hat{\boldsymbol{Y}}^{(t)}$. That is, we use the average set loss $\frac{1}{T} \sum_t L_{\text{set}}(\hat{\boldsymbol{Y}}^{(t)}, \boldsymbol{Y})$ as loss (similar to [4]). This

encourages $\hat{Y}$ to converge to $Y$ quickly and not diverge with more steps, which significantly increases the robustness of our algorithm.

We sometimes observed divergent training behaviour when the outer learning rate is set inappropriately. By replacing the instances of $||\cdot||^2$ in $L_{set}$ and $L_{repr}$ with the Huber loss (squared error for differences below 1 and absolute error above 1) – as is commonly done in object detection models – training became less sensitive to hyperparameter choices.

The inner optimisation can be modified to include a momentum term, which stops a prediction from oscillating around a solution. This gives us slightly better results, but we did not use this for any experiments to keep our method as simple as possible.

It is possible to explicitly include the sum of masks as a feature in the representation $z$ for our model. This improves our results on MNIST – likely due to the explicit signal for the model to predict the correct set size – but again, we do not use this for simplicity.

# 4    Related work

The main approach we compare our method to is the simple method of using an MLP decoder to predict sets. This has been used for predicting point clouds [1; 8], bounding boxes [20; 2], and graphs (sets of nodes and edges) [6; 22]. These predict an ordered representation (list) and treat it as if it is unordered (set). As we discussed in section 2, this approach runs into the responsibility problem. Some works on predicting 3d point clouds make domain-specific assumptions such as independence of points within a set [14] or grid-like structures [27]. To avoid inefficient graph matching losses, Yang et al. [26] compute a permutation-invariant loss between graphs by comparing them in the latent space (similar to our $L_{repr}$) in an adversarial setting.

An alternative approach is to use an RNN decoder to generate this list [15; 23; 25]. The problem can be made easier if it can be turned from a set into a sequence problem by giving a canonical order to the elements in the set through domain knowledge [25]. For example, You et al. [28] generate the nodes of a graph by ordering the set of nodes based on the traversal order of a breadth-first search.

The closest work to ours is by Mordatch [17]. They also iteratively minimise a function (their energy function) in each forward pass of the neural network and differentiate through the iteration to learn the weights. They have only demonstrated that this works for modifying small sets of 2d elements in relatively simple ways, so it is unclear whether their approach scales to the harder problems such as object detection that we tackle in this paper. In particular, minimising $L_{repr}$ in our model has the easy-to-understand consequence of making the predicted set more similar to the target set, while it is less clear what minimising their learned energy function $E(\hat{Y}, z)$ does.

Zhang et al. [30] construct an auto-encoder that pools a set into a feature vector where information from the encoder is shared with their decoder. This is done to make their decoder permutation-equivariant, which they use to avoid the responsibility problem. However, this strictly limits their decoder to usage in auto-encoders – not set prediction – because it requires an encoder to be present during inference.

Greff et al. [9] construct an auto-encoder for images with a set-structured latent space. They are able to find latent sets of variables to describe an image composed of a set of objects with some task-specific assumptions. While interesting from a representation learning perspective, our model is immediately useful in practice because it works for general supervised learning tasks.

Our inspiration for using backpropagation through an encoder as a decoder comes from the line of introspective neural networks [12; 13] for image modeling. An important difference is that in these works, the two optimisation loops (generating predictions and learning the network weights) are performed in sequence, while ours are nested. The nesting allows our outer optimisation to differentiate through the inner optimisation. This type of nested optimisation to obtain structured outputs with neural networks was first studied in [3; 4], of which our model can be considered an instance of. Note that [9] and [17] also differentiate through an optimisation, which suggests that this approach is of general benefit when working with sets. By differentiating through a decoder rather than an encoder, Bojanowski et al. [5] learn a representation instead of a prediction.

It is important to clearly separate the vector-to-set setting in this paper from some related works on set-to-set mappings, such as the equivariant version of Deep Sets [29] and self-attention [24]. Tasks

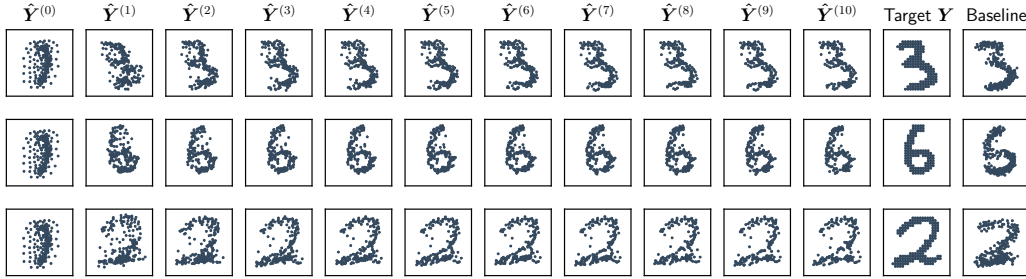

Figure 1: Progression of set prediction algorithm on MNIST ($\hat{Y}^{(t)}$). Our predictions come from our model with $0.08 \times 10^{-3}$ loss, while the baseline predictions come from an MLP decoder model with $0.09 \times 10^{-3}$ loss.

Table 1: Chamfer reconstruction loss on MNIST in thousandths. Lower is better. Mean and standard deviation over 6 runs.

| Model | Loss |
|---|---|
| MLP baseline | $0.21_{\pm 0.18}$ |
| RNN baseline | $0.49_{\pm 0.19}$ |
| Ours | $\mathbf{0.09}_{\pm 0.01}$ |

like object detection, where no set input is available, can not be solved with set-to-set methods alone; the feature vector from the image encoder has to be turned into a set first, for which a vector-to-set model like ours is necessary. Set-to-set methods do not have to deal with the responsibility problem, because the output usually has the same ordering as the input. Methods like [16] and [31] learn to predict a permutation matrix for a set (set-to-set-of-position-assignments). When this permutation is applied to the input set, the set is turned into a list (set-to-list). Again, our model is about producing a set as *output* while not necessarily taking a set as input.

## 5 Experiments

In the following experiments, we compare our set prediction network to a model that uses an MLP or RNN (LSTM) as set decoder. In all experiments, we fix the hyperparameters of our model to $T = 10, \eta = 800, \lambda = 0.1$. Further details about the model architectures, training settings, and hyperparameters are given in Appendix B. We provide the PyTorch [18] source code to reproduce all experiments at `https://github.com/Cyanogenoid/dspn`.

### 5.1 MNIST

We begin with the task of auto-encoding a set version of MNIST. A set is constructed from each image by including all the pixel coordinates (x and y, scaled to the interval $[0, 1]$) of pixels that have a value above the mean pixel value. The size of these sets varies from 32 to 342 across the dataset.

**Model**  In our model, we use a set encoder that processes each element individually with a 3-layer MLP, followed by FSPool [30] as pooling function to produce 256 latent variables. These are decoded with our algorithm to predict the input set. We compare this against a baseline model with the same encoder, but with a traditional MLP or LSTM as decoder. This approach to decoding sets is used in models such as in [1] (AE-CD variant) and [23]; these baselines are representative of the best approaches for set prediction in the literature. Note that these baselines have significantly more parameters than our model, since our decoder has almost no additional parameters by sharing the encoder weights (ours: $\sim$140 000 parameters, MLP: $\sim$530 000, LSTM: $\sim$470 000). For the baselines, we include a mask feature with each element to allow for variable-size sets. Due to the large maximum set size, use of Hungarian matching is too slow. Instead, we use the Chamfer loss to compute the loss between predicted and target set in this experiment.

Table 2: Average Precision (AP) for different intersection-over-union thresholds for a predicted bounding box to be considered correct. Higher is better. Mean and standard deviation over 6 runs.

| Model | $AP_{50}$ | $AP_{90}$ | $AP_{95}$ | $AP_{98}$ | $AP_{99}$ |
|---|---|---|---|---|---|
| MLP baseline | $99.3_{\pm 0.2}$ | $94.0_{\pm 1.9}$ | $57.9_{\pm 7.9}$ | $0.7_{\pm 0.2}$ | $0.0_{\pm 0.0}$ |
| RNN baseline | $99.4_{\pm 0.2}$ | $94.9_{\pm 2.0}$ | $65.0_{\pm 10.3}$ | $2.4_{\pm 0.0}$ | $0.0_{\pm 0.0}$ |
| Ours (10 iters) | $98.8_{\pm 0.3}$ | $94.3_{\pm 1.5}$ | $85.7_{\pm 3.0}$ | $\mathbf{34.5}_{\pm 5.7}$ | $\mathbf{2.9}_{\pm 1.2}$ |
| Ours (20 iters) | $\mathbf{99.8}_{\pm 0.0}$ | $\mathbf{98.7}_{\pm 1.1}$ | $\mathbf{86.2}_{\pm 7.2}$ | $24.3_{\pm 8.0}$ | $1.4_{\pm 0.9}$ |
| Ours (30 iters) | $\mathbf{99.8}_{\pm 0.1}$ | $96.7_{\pm 2.4}$ | $75.5_{\pm 12.3}$ | $17.4_{\pm 7.7}$ | $0.9_{\pm 0.7}$ |

**Results**    Table 1 shows that our model improves over the two baselines. In Figure 1, we show the progression of $\hat{Y}$ throughout the minimisation with $\hat{Y}^{(10)}$ as the final prediction, the ground-truth set, and the baseline prediction of an MLP decoder. Observe how every optimisation starts with the same set $\hat{Y}^{(0)}$, but is transformed differently depending on the gradient of $g_{enc}$. Through this minimisation of $L_{repr}$ by the inner optimisaton, the set is gradually changed into a shape that closely resembles the correct digit.

The types of errors of our model and the baseline are different, despite the use of models with similar losses in Figure 1. Errors in our model are mostly due to scattered points outside of the main shape of the digit, which is particularly visible in the third row. We believe that this is due to the limits of the encoder used: an encoder that is not powerful enough maps the slightly different sets to the same representation, so there is no $L_{repr}$ gradient to work with. It still models the general shape accurately, but misses the fine details of these scattered points. The MLP decoder has less of this scattering, but makes mistakes in the shape of the digit instead. For example, in the third row, the baseline has a different curve at the top and a shorter line at the bottom. This difference in types of errors is also present in the extended examples in Figure 3.

Note that reconstructions shown in [30] for the same auto-encoding task appear better because their decoder uses additional information outside of the latent space: they copy multiple $n \times n$ matrices from the encoder into the decoder. In contrast, all information about the set is completely contained in our permutation-invariant latent space.

## 5.2   Bounding box prediction

Next, we turn to the task of object detection on the CLEVR dataset [11], which contains 70,000 training and 15,000 validation images. The goal is to predict the set of bounding boxes for the objects in an image. The target set contains at most 10 elements with 4 dimensions each: the (normalised) x-y coordinates of the top-left and bottom-right corners of each box. As the dataset does not contain bounding box information canonically, we use [7] to calculate approximate bounding boxes. This causes the ground-truth bounding boxes to not always be perfect, which is a source of noise.

**Model**    We encode the image with a ResNet34 [10] into a 512d feature vector, which is fed into the set decoder. The set decoder predicts the set of bounding boxes *from this single feature vector* describing the whole image. This is in contrast to existing region proposal networks [19] for bounding box prediction where the use of the entire feature map is required for the typical anchor-based approach. As the set encoder in our model, we use a 2-layer relation network [21] with FSPool [30] as pooling. This is stronger than the FSPool-only model (without RN) we used in the MNIST experiment. We again compare this against a baseline that uses an MLP or LSTM as set decoder (matching AE-EMD [1] and [20] for the MLP decoder, [23] for the LSTM decoder). Since the sets are much smaller compared to our MNIST experiments, we can use the Hungarian loss as set loss. We perform no post-processing (such as non-maximum suppression) on the predictions of the model. The whole model is trained end-to-end.

**Results**    We show our results in Table 2 using the standard average precision (AP) metric used in object detection with sample predictions in Figure 2. Our model is able to very accurately localise the objects with high AP scores even when the intersection-over-union (IoU) threshold for a predicted box to match a groundtruth box is very strict. In particular, our model using 10 iterations (the same it was trained with) has much better $AP_{95}$ and $AP_{98}$ than the baselines. The shown baseline model

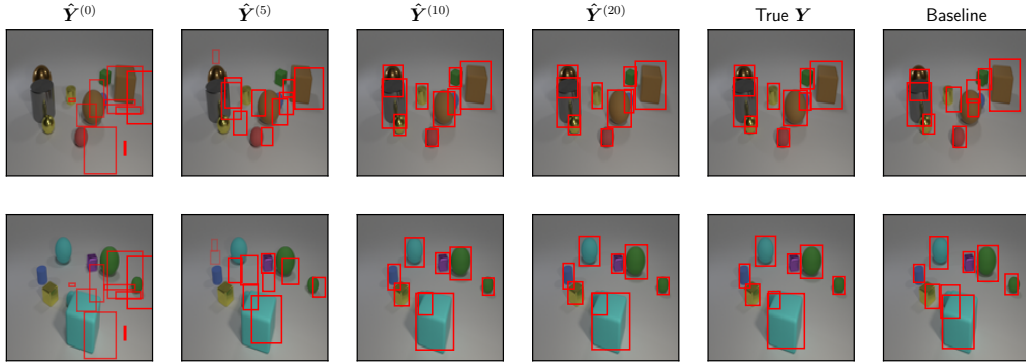

Figure 2: Progression of set prediction algorithm for bounding boxes in CLEVR. The shown MLP baseline sometimes struggles with heavily-overlapping objects and often fails to centre the object in the boxes.

can predict bounding boxes in the close vicinity of objects, but fails to place the bounding box precisely on the object. This is visible from the decent performance for low IoU thresholds, but bad performance for high IoU thresholds.

We can also run our model with more inner optimisation steps than the 10 it was trained with. Many results improve when doubling the number of steps, which shows that further minimisation of $L_{\text{repr}}(\hat{Y}, z)$ is still beneficial, even if it is unseen during training. The model "knows" that its prediction is still suboptimal when $L_{\text{repr}}$ is high and also how to change the set to decrease it. This confirms that the optimisation is reasonably stable and does not diverge significantly with more steps. Being able to change the number of steps allows for a dynamic trade-off between prediction quality and inference time depending on what is needed for a given task.

The less-strict AP metrics (which measure large mistakes) improve with more iterations, while the very strict $AP_{98}$ and $AP_{99}$ metrics consistently worsen. This is a sign that the inner optimisation learned to reach its best prediction at exactly 10 steps, but slightly overshoots when run for longer. The model has learned that it does not fully converge with 10 steps, so it is compensating for that by slightly biasing the inner optimisation to get a better 10 step prediction. This is at the expense of the strictest AP metrics worsening with 20 steps, where this bias is not necessary anymore.

Bear in mind that we do not intend to directly compete against traditional object detection methods. Our goal is to demonstrate that our model can accurately predict a set from a single feature vector, which is of general use for set prediction tasks not limited to image inputs.

### 5.3 State prediction

Lastly, we want to directly predict the full state of a scene from images on CLEVR. This is the set of objects with their position in the 3d scene (x, y, z coordinates), shape (sphere, cylinder, cube), colour (eight colours), size (small, large), and material (metal/shiny, rubber/matte) as features. For example, an object can be a "small cyan metal cube" at position (0.95, -2.83, 0.35). We encode the categorial features as one-hot vectors and concatenate them into an 18d feature vector for each object. Note that we do not use bounding box information, so the model has to implicitly learn which object in the image corresponds to which set element with the associated properties. This makes it different from usual object detection tasks, since bounding boxes are required for traditional object detection models that rely on anchors.

**Model**  We use exactly the same model as for the bounding box prediction in the previous experiment with all hyperparameters kept the same. The only difference is that it now outputs 18d instead of 4d set elements. For simplicity, we continue using the Hungarian loss with the Huber loss as pairwise cost, as opposed to switching to cross-entropy for the categorical features.

**Results**  We show our results in Table 3 and give sample outputs in Appendix C. The evaluation metric is the standard average precision as used in object detection, with the modification that

Table 3: Average Precision (AP) in % for different distance thresholds of a predicted set element to be considered correct. $AP_\infty$ only requires all attributes to be correct, regardless of 3d position. Higher is better. Mean and standard deviation over 6 runs.

| Model | $AP_\infty$ | $AP_1$ | $AP_{0.5}$ | $AP_{0.25}$ | $AP_{0.125}$ |
|---|---|---|---|---|---|
| MLP baseline | $3.6_{\pm0.5}$ | $1.5_{\pm0.4}$ | $0.8_{\pm0.3}$ | $0.2_{\pm0.1}$ | $0.0_{\pm0.0}$ |
| RNN baseline | $4.0_{\pm1.9}$ | $1.8_{\pm1.2}$ | $0.9_{\pm0.5}$ | $0.2_{\pm0.1}$ | $0.0_{\pm0.0}$ |
| Ours (10 iters) | $72.8_{\pm2.3}$ | $59.2_{\pm2.8}$ | $39.0_{\pm4.4}$ | $12.4_{\pm2.5}$ | $1.3_{\pm0.4}$ |
| Ours (20 iters) | $84.0_{\pm4.5}$ | $80.0_{\pm4.9}$ | $\mathbf{57.0}_{\pm12.1}$ | $\mathbf{16.6}_{\pm9.0}$ | $\mathbf{1.6}_{\pm0.9}$ |
| Ours (30 iters) | $\mathbf{85.2}_{\pm4.8}$ | $\mathbf{81.1}_{\pm5.2}$ | $47.4_{\pm17.6}$ | $10.8_{\pm9.0}$ | $0.6_{\pm0.7}$ |

a prediction is considered correct if there is a matching groundtruth object with exactly the same properties and within a given Euclidean distance of the 3d coordinates. Our model clearly outperforms the baselines. This shows that our model is also suitable for modeling high-dimensional set elements.

When evaluating with more steps than our model was trained with, the difference in the more lenient metrics improves even up to 30 iterations. This time, the results for 20 iterations are all better than for 10 iterations. This suggests that 10 steps is too few to reach a good solution in training, likely due to the higher difficulty of this task compared to the bounding box prediction. Still, the representation $z$ that the input encoder produces is good enough such that minimising $L_{\text{repr}}$ more at evaluation time leads to better results. When going up to 30 iterations, the result for predicting the state only (excluding 3d position) improves further, but the accuracy of the 3d position worsens. We believe that this is again caused by overshooting the target due to the bias of training the model with only 10 iterations.

## 6 Discussion

In this paper we showed how to predict sets with a deep neural network in a way that respects the set structure of the problem. We demonstrated in our experiments that this works for small (size 10) and large sets (up to size 342), as well as low-dimensional (2d) and higher-dimensional (18d) set elements. Our model is consistently better than the baselines across all experiments by predicting sets properly, rather than predicting a list and pretending that it is a set.

The improved results of our approach come at a higher computational cost. Each evaluation of the network requires time for $O(T)$ passes through the set encoder, which makes training take about 75% longer on CLEVR with $T = 10$. Keep in mind that this only involves the set encoder (which can be fairly small), not the input encoder (such as a CNN or RNN) that produces the target $z$. Further study into representationally-powerful and efficient set encoders such as RN [21] and FSPool [30] – which we found to be critical for good results in our experiments – would be of considerable interest, as it could speed up the convergence and thus inference time of our method. Another promising approach is to better initialise $Y^{(0)}$ – perhaps with an MLP – so that the set needs to be changed less to minimise $L_{\text{repr}}$. Our model would act as a set-aware refinement method of the MLP prediction. Lastly, stopping criteria other than iterating for a fixed 10 steps can be used, such as stopping when $L_{\text{repr}}(g_{\text{enc}}(\hat{Y}), z)$ is below a fixed threshold: this would stop when the encoder thinks $\hat{Y}$ is of a certain quality corresponding to that threshold.

Our algorithm may be suitable for generating samples under other invariance properties. For example, we may want to generate images of objects where the rotation of the object does not matter (such as aerial images). Using our decoding algorithm with a rotation-invariant image encoder could predict images without forcing the model to choose a fixed orientation of the image, which could be a useful inductive bias.

In conclusion, we are excited about enabling a wider variety of set prediction problems to be tackled with deep neural networks. Our main idea should be readily extensible to similar domains such as graphs to allow for better graph prediction, for example molecular graph generation or end-to-end scene graph prediction from images. We hope that our model inspires further research into graph generation, stronger object detection models, and – more generally – a more principled approach to set prediction.

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
