[Supplementary Material]

# A  Proof of permutation-equivariance

**Definition 1.** *A function $f : \mathbb{R}^{n \times c} \to \mathbb{R}^d$ is permutation-invariant iff it satisfies:*

$$f(\boldsymbol{X}) = f(\boldsymbol{PX}) \tag{7}$$

*for all permutation matrices $\boldsymbol{P}$.*

**Definition 2.** *A function $g : \mathbb{R}^{n \times c} \to \mathbb{R}^{n \times d}$ is permutation-equivariant iff it satisfies:*

$$\boldsymbol{P}g(\boldsymbol{X}) = g(\boldsymbol{PX}) \tag{8}$$

*for all permutation matrices $\boldsymbol{P}$.*

**Theorem 1.** *The gradient of a permutation-invariant function $f : \mathbb{R}^{n \times c} \to \mathbb{R}^d$ with respect to its input is permutation-equivariant:*

$$\boldsymbol{P}\frac{\partial f(\boldsymbol{X})}{\partial \boldsymbol{X}} = \frac{\partial f(\boldsymbol{PX})}{\partial \boldsymbol{PX}} \tag{9}$$

*Proof.* Using Definition 1, the chain rule, and the orthogonality of $\boldsymbol{P}$:

$$\boldsymbol{P}\frac{\partial f(\boldsymbol{X})}{\partial \boldsymbol{X}} = \boldsymbol{P}\frac{\partial f(\boldsymbol{PX})}{\partial \boldsymbol{X}} \tag{10}$$

$$= \boldsymbol{P}\frac{\partial \boldsymbol{PX}}{\partial \boldsymbol{X}}\frac{\partial f(\boldsymbol{PX})}{\partial \boldsymbol{PX}} \tag{11}$$

$$= \boldsymbol{PP}^T\frac{\partial f(\boldsymbol{PX})}{\partial \boldsymbol{PX}} \tag{12}$$

$$= \frac{\partial f(\boldsymbol{PX})}{\partial \boldsymbol{PX}} \tag{13}$$

$\square$

# B  Details

In our algorithm, $\eta$ was chosen in initial experiments and we did not tune it beyond that. We did this by increasing $\eta$ until the output set visibly changed between inner optimisation steps when the set encoder is randomly initialised. This makes it so that changing the set encoder weights has a noticeable effect rather than being stuck with $\hat{\boldsymbol{Y}}^{(T)} \approx \hat{\boldsymbol{Y}}^{(0)}$.

$T = 10$ was chosen because it seemed to be enough to converge to good solutions on MNIST. We simply kept this for the supervised experiments on CLEVR.

In the supervised experiments, we would often observe large spikes in training that cause the model diverge when $\lambda = 1$. By changing around various parameters, we found that reducing $\lambda$ eliminated most of this issue and also made training converge to better solutions. Much smaller values than 0.1 converged to worse solutions. This is likely because the issue of not having the $L_{\text{repr}}(\boldsymbol{Y}, \boldsymbol{z})$ term in the outer loss in the first place ($\lambda = 0$) is present again – see subsection 3.2.

For all experiments, we used Adam with the default momentum values and batch size 32 for the outer optimisation. The only hyperparameter we tuned in the experiments is the learning rate of the outer optimisation. Every individual experiment is run on a single 1080 Ti GPU.

The MLP decoder baseline has 3 layers with 256 (MNIST) or 512 (CLEVR) neurons in the first two layers and the number of channels of the output set in the task in the third layer. The LSTM decoder linearly transforms the latent space into 256 (MNIST) or 512 (CLEVR) dimensions, which is used as initial cell state of the LSTM. The LSTM is run for the same number of steps as the maximum set size, and the outputs of these steps is each linearly transformed into the output dimensionality.

## B.1  MNIST

For MNIST, we train our model and the baseline model for 100 epochs to make sure that they have converged. Both models have a 3-layer MLP with ReLU activations and 256 neurons in the three layers. For simplicity, sets are padded to a fixed size for FSPool. FSPool has 20 pieces in its piecewise linear

function. We tried learning rates in $\{1.0, 0.1, 0.03, 0.01, 0.003, 0.001, 0.0003, 0.0001, 0.00001\}$ and chose 0.01. For the baselines, none of the other learning rates performed significantly better than the one we chose.

The baselines are trained slightly differently to our model. They do not output mask values natively, so we have to train them with the mask values in the training target. In other words, they are trained to predict x coordinate, y coordinate, and the mask for each point. We found it crucial to explicitly add 1 to the mask in the baseline model for good results. Otherwise, many of the baseline outputs get stuck in the local optimum of predicting the (0, 0, 0) point and the output is too sparse.

## B.2   CLEVR

We train our model and the baselines models for 100 epochs on the training set of CLEVR and evaluate on the validation set, since no ground-truth scene information is available for the test set. All images are resized to 128×128 resolution. The set encoder is a 2-layer Relation Network with ReLU activation between the two layers, wherein the sum pooling is replaced with FSPool. The two layers have 512 neurons each. Because we use the Hungarian loss instead of the Chamfer loss here, including the mask feature in the target set does not worsen results, so we include the mask target for both the baseline and our model for consistency. To tune the learning rate, we started with the learning rate found for MNIST and decreased it similarly-sized steps until the training accuracy after 100 epochs worsened. We settled on 0.0003 as learning rate for both the bounding box and the state prediction task. All other hyperparameters are kept the same as for MNIST. The ResNet34 that encodes the image is not pre-trained.

# C  Additional outputs

Figure 3: Progression of set prediction algorithm on MNIST.

Figure 4: Progression of set prediction algorithm on CLEVR bounding boxes.

Table 4: Progression of set prediction algorithm on CLEVR state prediction. Red text denotes a wrong attribute. Objects are sorted by x coordinate, so they are sometimes misaligned with wrongly-coloured red text (see third example: red entries in $\hat{Y}^{(20)}$ and bottom two red entries in baseline).

| $\hat{Y}^{(5)}$ | $\hat{Y}^{(10)}$ | $\hat{Y}^{(20)}$ | True $Y$ | Baseline |
|---|---|---|---|---|
| (-0.14, 1.16, 3.57) large purple rubber sphere | (-2.33, -2.41, 0.73) large yellow metal cube | (-2.33, -2.42, 0.78) large yellow metal cube | (-2.42, -2.40, 0.70) large yellow metal cube | (-1.65, -2.85, 0.69) large yellow metal cube |
| (0.01, 0.12, 3.42) large gray metal cube | (-1.20, 1.27, 0.67) large purple rubber sphere | (-1.21, 1.20, 0.65) large purple rubber sphere | (-1.18, 1.25, 0.70) large purple rubber sphere | (-0.95, 1.08, 0.68) large green rubber sphere |
| (0.67, 0.65, 3.38) small purple metal cube | (-0.96, 2.54, 0.36) small gray rubber sphere | (-0.96, 2.59, 0.36) small gray rubber sphere | (-1.02, 2.61, 0.35) small gray rubber sphere | (-0.40, 2.14, 0.35) small red rubber sphere |
| (0.67, 1.14, 2.96) small purple rubber sphere | (1.61, 1.57, 0.36) small yellow metal cube | (1.58, 1.62, 0.38) small purple metal cube | (1.74, 1.53, 0.35) small purple metal cube | (1.68, 1.77, 0.35) small brown metal cube |

| $\hat{Y}^{(5)}$ | $\hat{Y}^{(10)}$ | $\hat{Y}^{(20)}$ | True $Y$ | Baseline |
|---|---|---|---|---|
| (-0.29, 1.14, 3.73) small purple metal cube | (-2.78, 0.86, 0.72) large cyan rubber sphere | (-2.62, 0.83, 0.68) large cyan rubber sphere | (-2.88, 0.78, 0.70) large cyan rubber sphere | (-2.42, 0.63, 0.71) large purple rubber sphere |
| (-0.11, -0.37, 3.65) small brown metal cube | (-2.17, -1.59, 0.38) small blue rubber cylinder | (-2.12, -1.58, 0.49) small blue rubber cylinder | (-2.14, -1.63, 0.35) small blue rubber cylinder | (-2.40, -2.07, 0.35) small green rubber cylinder |
| (0.08, 0.56, 3.84) large cyan rubber cube | (-0.45, 2.19, 0.40) small purple metal cube | (-0.60, 2.23, 0.29) small purple metal cube | (-0.78, 1.97, 0.35) small purple metal cube | (-0.74, 2.46, 0.33) small cyan metal cube |
| (0.69, -0.43, 3.55) small brown rubber sphere | (-0.14, -2.15, 0.38) small yellow metal cube | (-0.30, -1.99, 0.32) small yellow metal cube | (-0.38, -2.06, 0.35) small yellow metal cube | (0.30, -1.86, 0.34) small gray rubber sphere |
| (1.12, 0.21, 3.83) large cyan rubber cube | (0.53, 2.56, 0.70) large green rubber sphere | (0.27, 2.46, 0.72) large green rubber sphere | (0.42, 2.56, 0.70) large green rubber sphere | (0.69, -2.10, 0.36) small red metal cube |
| (1.23, -0.25, 3.58) small cyan rubber sphere | (0.93, -1.41, 0.35) small cyan rubber sphere | (0.86, -1.31, 0.27) small cyan rubber sphere | (0.81, -1.30, 0.35) small cyan rubber sphere | (1.12, 2.28, 0.70) large cyan rubber sphere |
| (1.73, 1.04, 3.57) small cyan rubber sphere | (2.50, -2.08, 0.76) large cyan rubber cube | (2.64, -2.05, 0.76) large cyan rubber cube | (2.56, -1.94, 0.70) large cyan rubber cube | (2.55, -2.26, 0.73) large yellow rubber cube |
| (2.06, 1.94, 3.81) large brown rubber sphere | (2.61, 2.59, 0.33) small green rubber sphere | (2.75, 2.73, 0.35) small green rubber sphere | (2.74, 2.64, 0.35) small green rubber sphere | (2.99, 2.59, 0.35) small purple rubber sphere |

| $\hat{Y}^{(5)}$ | $\hat{Y}^{(10)}$ | $\hat{Y}^{(20)}$ | True $Y$ | Baseline |
|---|---|---|---|---|
| (0.22, 0.12, 3.47) small brown rubber cube | (-2.76, -1.42, 0.68) large blue metal cylinder | (-2.68, -1.64, 0.77) large blue metal cylinder | (-2.62, -1.76, 0.70) large blue metal cylinder | (-2.47, -1.73, 0.70) large cyan metal cylinder |
| (0.41, 0.11, 3.77) large gray metal cube | (-1.56, -0.61, 0.35) small blue rubber cylinder | (-2.43, 0.03, 0.34) small blue rubber cube | (-2.29, 0.49, 0.35) small blue rubber cube | (-2.42, 0.09, 0.36) small blue rubber cylinder |
| (0.50, 0.44, 3.61) small gray rubber cube | (-1.08, 0.23, 0.33) small green rubber cube | (-1.00, 1.18, 0.33) small red rubber cylinder | (-0.93, 1.15, 0.35) small red rubber cylinder | (-1.24, 1.16, 0.36) small red rubber cube |
| (0.83, 0.53, 3.45) small cyan rubber sphere | (-0.07, 0.97, 0.36) small green rubber cylinder | (-0.01, -1.00, 0.46) small green rubber cube | (0.28, -2.84, 0.35) small cyan rubber cylinder | (0.39, 0.20, 0.33) small red rubber sphere |
| (0.86, 0.85, 3.50) small gray rubber sphere | (0.28, -2.44, 0.49) small cyan rubber cylinder | (0.21, -2.88, 0.40) small cyan rubber cylinder | (0.29, -0.98, 0.35) small green rubber cube | (0.56, -3.11, 0.35) small yellow rubber cylinder |
| (1.86, 2.34, 3.80) large gray metal cube | (1.36, -0.63, 0.38) small cyan rubber cylinder | (0.99, 0.17, 0.37) small cyan rubber cylinder | (0.92, 0.54, 0.35) small green rubber cube | (0.90, 0.64, 0.35) small green rubber sphere |
| (1.97, 0.55, 3.61) small green rubber sphere | (2.01, 3.07, 0.65) small green rubber sphere | (1.97, 2.89, 0.39) small green rubber sphere | (2.04, 2.78, 0.70) small green rubber sphere | (2.39, 0.27, 0.36) small green rubber sphere |
| | (2.69, 0.63, 0.34) large gray metal cube | (2.87, 0.51, 0.25) large gray metal cube | (2.70, 0.67, 0.35) large gray metal cube | (2.44, 2.55, 0.68) small yellow rubber sphere |
| | small yellow rubber sphere | small yellow rubber sphere | small yellow rubber sphere | large gray metal cube |