[Reviews · NeurIPS 2019]

Reviewer 1



The work is fairly interesting. It addresses set prediction that has been frequently encountered in a range of problems such as object detection where the order-invariant of a collection of predictions is preferred. It is particularly inspiring that the authors leverages the property of an encoding process that is order invariant and reverses the process for set prediction. In the authors' own words, "to decode a feature vector into a set, we can use gradient descent to find a set that encodes to that feature vector." I have a few questions about the model and the experiments: 1. For the auto-encoder case, "the same set encoder is used in encoding and decoding". Does it mean the bottleneck, z, is of the same dimension as Y, (dxn)? 2. For Algorithm 8, is the loss on line 8 for training the set decoder? If so, this should be made clear in the formulation, so is in Equation 6. 3. The paper mentions that the two optimization processes are nested. However, they seem to be performed in seq from Algorithm 1. Please clarify it. 4. Did the baseline models with MLP use Equation 1 and 2? MLP would benefit from Chamfer and Hungarian matching as well, which I feel is what are typically used in existing solutions. To me, Line 2-7 in Algorithm 1 is a core part of the contribution of the paper, rather than the Eq. 1 and 2. =========== Thanks for your rebuttal that answers my questions. My rating towards the paper remains positive.

Reviewer 2



Summary: This paper presents an approach for solving machine learning tasks that require the prediction to be presented in the form of a set. The authors propose to use the set encoder (which is composed of permutation-invariant operations) at the prediction phase by finding an output set with an optimization procedure. As the model output is a vector of continuous features for each set element, it can be done by means of nested gradient descent optimization. In order to solve the task of set prediction for external feature vector, the work suggests a combined loss function that encourages the representation of ground truth to be close to obtained features. Results are shown on MNIST and CLEVR datasets and outperform those of an MLP baseline. The proposed approach is a promising new direction in the area of set prediction, although the choice of baselines makes the experimental setup less convincing. Quality: The paper argues that ignoring the structure might yield suboptimal results for the task of set prediction due to the responsibility issue, illustrating that with a simple example. The experimental setup, however, is not entirely convincing regarding the demonstration of method superiority. In particular, 1. For the experiment on MNIST there are no quantitative results to evaluate the quality of both approaches; if no generally accepted metrics for this kind of task are known, it should at least be possible to use Chamfer loss or Intersection-over-Union score for comparison. 2. Secondly, the baseline approach seemingly underperforms by a large margin in all experiments, especially in the MNIST autoencoding task, thus rendering the claims of superior performance as compared with existing methods less sound. It would help the paper if the authors provided results of previously existing methods more suitable to the task of set prediction, such as simple sequence-to-sequence models (even if the order is arbitrary) based on recurrence or self-attention or approaches like (Rezatofighi17). 3. For the MLP baseline on MNIST dataset, it appears that the main issue resides in the inability of the model to predict a discrete mask with high quality; it might be useful to compare both methods on the task of fixed cardinality set generation. 4. Additionally, although the paper uses FSPool for encoding the input set, there is no comparison with FSUnpool, which was proposed in the same work as FSPool for decoding a set from its representation. Furthermore, sorting the features in the encoder and not restoring the order in the decoder can be harmful for the training process; can that be one of the reasons for poor performance of the baseline model? Originality: To the best of my knowledge, the method of predicting a set with the use of an encoder and iterative optimization is a fairly novel idea. The authors also provide proof of permutation invariance of the gradient of any permutation-invariant function, which was not shown before in relevant works on the subject. Clarity: Overall, the paper is well-written and clear; the authors provide a detailed description of the experiments which should be enough to reproduce the results. Significance: This work provides a unique approach to the problem of set prediction for autoencoding as well as generating a set from external representations, such as those obtained by CNNs on images. Right now it is not entirely evident whether the idea works better than currently known methods in practice, but I expect a more thorough experimental comparison to clarify that. (Rezatofighi17) Rezatofighi, S. Hamid and G, Vijay Kumar B. and Milan, Anton and Abbasnejad, Ehsan and Dick, Anthony and Reid, Ian. DeepSetNet: Predicting Sets with Deep Neural Networks. ICCV 2017. --------------------------------------------------------------------------------------------- I would like to thank the authors for addressing my comments. The answers to most questions are clear and give enough details for a better understanding of the approach proposed in the paper. Provided that the authors add the stated experimental results (metrics on MNIST dataset, results of LSTM baseline on all datasets, ablation study with poolings other than FSPool) and address all questions raised by the reviewers in the final version of the paper, I will raise the score of this submission. I would especially like to see the quantitative results on MNIST for the fixed size set prediction task in the final version. Even though it is simpler than the problem considered in the paper, currently the MLP baseline seems unsuitable for variable size set prediction, which harms the credibility of this experiment.

Reviewer 3



Originality: Novel. The proposed approach is novel in utilizing the idea of output space gradient descent in making sure it is permutation-invariant. Although similar ideas has been adopted in some other scenarios such as other generative model problems, this idea has not yet been formally generalized and applied to set prediction problems. So I believe this paper has sufficient novelty. Quality: High quality. 1. The paper solves an important problem which hasn't been well studied by the community. Set prediction is a very interesting problem which has lots of potential applications such as object detection, relationship detection, etc. 2. The paper is very well written. The experimental validation is convincing which shows that the proposed method is promising. However, more set prediction baselines would be even helpful. Clarity: Very clear. The paper is very well written and clear. It is a pleasure to read the paper. I think the paper very carefully organized and structured which can keep a reader’s interests throughout the whole paper. Significance: Significant and inspiring. I believe the paper has great significance. The two major ideas can potentially inspire a lot of researchers in this area: (1) the idea of conducting optimization over the output space to achieve a task and (2) the idea of gradient descent in the output space keeps permutation invariance. The experiments prove such idea is promising although they can be further improved. But I think beating the state-of-the-art is not the main goal of this paper given its other intellectual contribution.

Reviewer 4



The authors propose a methodology for training deep neural networks to predict on discrete sets of objects. This can be useful for a variety of tasks such as matching or sorting. The authors accomplish this by mapping from a latent representation (z) to a set (Y) by incorporating a representation loss (for representing the set as z) that is iteratively refined through gradient descent. The paper is well written and easy to follow. The proposed method seems reasonable and simple to implement. However, the authors do not compare to any existing literature on deep learning for sets including [20], [23] and Mena et al. (Learning Latent Permutations with Gumbel Sinkhorn Networks). How does this work improve over those methods? In particular, Mena et al. learn a mapping from arbitrary feature vectors to a latent representation over sets using deep networks that is then decoded into a permutation. This would seem like a much more reasonable baseline than the simple feedforward network provided in the experiments. In Section 5.3, Mena et al. conduct an experiment to map scrambled pixels from MNIST digits into position assignments to reconstruct the digits. This seems like a very similar experiment to the first one of this paper. As such, why isn't that work compared to? Why can't [20] or [23] be applied to this problem as well, to provide stronger baselines? The authors repeatedly reference (5 times) an unpublished paper [2] (that is presumably from the authors) as well for motivation and comparison. It seems somewhat dubious to self-cite an un-peer reviewed article provided in the supplementary since it abuses the page limit. The reviewers need to either assume it is correct and backs up the citations or they need to review another additional paper to verify. In all the experiments, it seems like at some point running more iterations of the inner loop causes significantly worse results. Is this due to some kind of overfitting? Is there some sort of regularization that can be done here to make the algorithm more robust. Is there some guidance as to how this can be overcome?

[Author Response · NeurIPS 2019]

## Reviewer 1

**1.** No, they are usually different. The set encoder always maps its $d \times n$ input to a z-space. It is used to encode the ground-truth input set (use as encoder) and encode all the $\hat{Y}^{(t)}$ sets (use in decoder).

**2.** Yes, this is the loss for the outer optimisation to train the network weights. We will try to make this clearer.

**3.** They are nested in the sense that differentiating the outer loss requires differentiating through the inner optimisation procedure too. The inner optimisation is nested within the outer optimisation. Only one step of the outer optimisation is shown, so there is an implicit `for` loop around Algorithm 1 for training. We will try to make this clearer too.

**4.** The MLP baseline uses Eq. 1 and 2, just like in existing work that uses MLPs to predict sets. Algorithm 1 is indeed a core part of our contribution. It is placed near the main model section, while Eq. 1 and 2 are firmly in the background section. We will update the introduction to explicitly mention that Algorithm 1 is part of our core contribution.

## Reviewer 2

Correction: our proof is that the gradient of a permutation-invariant function is permutation-*equivariant*, not permutation-*invariant* as you stated in your review. There is a small but important difference.

**1.** The results of the MLP baseline are obviously so much worse that we didn't feel the need to include this. They are what you could expect from the difference in qualitative results and we can include them in an update. In particular, the test Chamfer loss is 0.00010 for our model and 0.00076 for the MLP on MNIST with mask feature.

**2.** The MLP baseline, while simple, *is exactly what many existing papers use* for set prediction (see the first sentence in the related works section for some of them). The MLP is the most suitable baseline because it is both simple (no need for much extra explanation for a reader to understand), as well as actually widely-used for set prediction in the literature. We explain why MLPs should struggle with modeling sets, so it shouldn't be a surprise when they do struggle with sets in our experiments. The only other relevant approach is RNN-based, which also has the responsibility problem.

We tried an LSTM (like in [19]) as set decoder, which improves the MNIST baseline to 0.00023 loss and is similar to the MLP on CLEVR (e.g. bbox $AP_{95} = 61.2\%$, state $AP_{\infty} = 3.3\%$), so still much worse than our model. Self-attention maps sets to sets, so it can only be used once you already have a set, not for vec-to-set as we are dealing with. Rezatofighi2017 is irrelevant (their title is not as general as it claims to be) because it's more about multi-labeling tasks rather than general set prediction: their sets are sets of discrete labels from a small finite domain, so they can be ordered just like in normal classification or multi-label tasks and is thus "easy". Our work deals with sets where the elements are feature vectors of real numbers, which their method doesn't apply to. In Rezatofighi's follow-up paper (we cite this as [16]) using the same problem set-up as our paper, they use a Hungarian loss and an MLP decoder. So, by comparing against an MLP baseline, we *are* comparing against their (and others') work [1]. We will make this more explicit.

**3.** The vast majority of sets in the real world are variable size, so the case of fixed size is not particularly relevant. We follow [16] with our MLP baseline, which has this exact mask output. Removing it seems to improve results of the MLP on MNIST to losses similar to our model, but now it is solving an easier task than our model is intended for.

**4.** FSUnpool is irrelevant because it can only be applied in auto-encoders, not in general set prediction. We state this in Lines 182–183, 223–226. We are evaluating set prediction methods on auto-encoding, not set auto-encoders specifically. Lines 217–218 state that replacing FSPool with other poolings (we tried mean, max, sum) makes the baseline worse.

## Reviewer 3

**1.** Instead of stopping at a fixed 10 iterations as we did, an alternative is to stop when $L_{\text{repr}}(g_{\text{enc}}(\hat{Y}), z)$ is below some threshold: this stops when the encoder thinks the representation is of a certain quality corresponding to that threshold. A lower $L_{\text{repr}}$ means a higher quality prediction because the representation of prediction and target match better.

**2.** When computing the outer loss, the initial state $\hat{Y}^{(0)}$ is learned too, which should help with the initialisation. In essence, $\hat{Y}^{(0)}$ is moved to a state that is close to training examples in set space on average, which reduces the distance the inner optimisation has to travel. This is balanced with a $\hat{Y}^{(0)}$ that is close in the representation space; this can be learned because the inner optimisation is being differentiated through.

**3.** Please refer to our answer to Reviewer 2, point 2, for an additional LSTM baseline.

## Footnotes

[1] The other things they developed (permutation head $O_2$, cardinality head $\alpha$) are not actually used (they stated this in their openreview rebuttal) or don't help in the experiments in their main body respectively. Thus, their approach is almost exactly the same as our MLP baseline and their experiment in section 4.1 is very similar to how we used the MLP in section 5.2 for bounding boxes.


[Meta-Review · NeurIPS 2019]

This submission deals with a fundamental problems in ML and presents an interesting approach to permutation invariant learning. The ideas of this work are interesting despite the weak baseline used in the experiments. The authors are requested to include stronger baselines in the final version.